# Disentangled deep generative models reveal coding principles of the human face processing network

**Paul Soulos, Leyla Isik** *

Department of Cognitive Science, Johns Hopkins University, Baltimore, Maryland, United States of America

* lisik@jhu.edu

**Data Availability Statement:** All data used were publicly available and linked to the original paper: https://openneuro.org/datasets/ds001761. All analysis code is publicly available on our github repo https://github.com/psoulos/disentangle-faces.

## Abstract

Despite decades of research, much is still unknown about the computations carried out in the human face processing network. Recently, deep networks have been proposed as a computational account of human visual processing, but while they provide a good match to neural data throughout visual cortex, they lack interpretability. We introduce a method for interpreting brain activity using a new class of deep generative models, disentangled representation learning models, which learn a low-dimensional latent space that "disentangles" different semantically meaningful dimensions of faces, such as rotation, lighting, or hairstyle, in an unsupervised manner by enforcing statistical independence between dimensions. We find that the majority of our model's learned latent dimensions are interpretable by human raters. Further, these latent dimensions serve as a good encoding model for human fMRI data. We next investigate the representation of different latent dimensions across face-selective voxels. We find that low- and high-level face features are represented in posterior and anterior face-selective regions, respectively, corroborating prior models of human face recognition. Interestingly, though, we find identity-relevant and irrelevant face features across the face processing network. Finally, we provide new insight into the few "entangled" (uninterpretable) dimensions in our model by showing that they match responses in the ventral stream and carry information about facial identity. Disentangled face encoding models provide an exciting alternative to standard "black box" deep learning approaches for modeling and interpreting human brain data.

## Author summary

We use a class of interpretable deep neural network models, disentangled variational autoencoders (dVAEs), to analyze human fMRI data. We find that a dVAE learns human interpretable dimensions of faces, such as lighting, expression, and hairstyle, and provides as good a match to human fMRI data as matched, non-disentangled models. Our disentangled encoding approach allows us to map different disentangled features to ROI and voxel activity. A decoding analysis confirms that the model separates identity relevant and irrelevant information and reveals that the remaining entangled dimensions contain

**Funding:** This work was supported with funds from The Clare Boothe Luce Program for Women in STEM (LI). The funders had no role in study design, data collection and analysis, decision to publish, or preparation of the manuscript.

**Competing interests:** I have read the journal's policy and the authors of this manuscript have the following competing interests: LI previously served on the editorial board of PLOS Computational Biology (2019-2022).

identity-relevant information. Together these results highlight the use of disentangled models for more interpretable fMRI encoding than standard deep learning models.

## Introduction

Humans are highly skilled at recognizing faces despite the complex high dimensional space that face stimuli occupy and the many transformations they undergo. Some dimensions (such as 3D rotation and lighting) are constantly changing and thus irrelevant to recognizing a face, while others (such as facial features or skin tone) are generally stable and useful for recognizing an individual's identity, and still others (such as hairstyle) can change but also offer important clues to identity. Face processing networks in the macaque and human brain have been thoroughly mapped [1–3] and many general coding principles have been identified, including separation of static vs. dynamic face representations [4,5] and increasing transformation invariance from posterior to anterior regions [6]. However, much is still unknown about the computations carried out across these regions, particularly in the human brain. Even fundamental information, such as how facial identity is represented, is still largely unknown [2]. This lack of understanding can be seen in the relatively poor decoding of face identity from fMRI data compared to other visual categories [7].

Recently, deep convolutional neural networks (DCNNs) trained on face recognition have been shown to learn effective face representations that provide a good match to human behavior [8], but such discriminatively trained models are difficult to interpret [9] and provide a poor match to human neural data [10]. Alternatively, deep generative models have been shown to provide a good match to human fMRI face processing data [11]. These models, however, transform faces into complex high dimensional latent spaces and thus suffer from the same lack of interpretability as standard DCNNs. Here we use a new class of deep generative models, disentangled representation learning models that isolate semantically meaningful factors of variation in individual latent dimensions, to understand the neural computations underlying human face processing.

Multiple disentangled representation learning models have been developed [12–17], many of which are based on Variational Autoencoders (VAEs) [18]. These disentangled variational autoencoders (dVAEs) learn a latent space that "disentangles" different explanatory factors in the training distribution by enforcing statistical independence between latent dimensions during training [19]. Intriguingly, when applied to faces, dVAEs have been shown to learn latent dimensions that are not only statistically independent, but also isolate specific, interpretable face features.

dVAEs learn a latent representation that is compact and highly interpretable by humans, so we investigate complex face representations across the human brain using dVAEs in an encoding model framework. We find that representations in disentangled models match those found in the human face processing network at least as well or better than standard deep learning models without the disentanglement cost (including traditional VAEs and DCNNs). We then map the learned semantically meaningful dVAE dimensions to voxel responses and quantify their facial identity information, providing new insight into the models and the human face processing network.

## Results

### Disentangled generative models factor latent space into human-interpretable dimensions

We trained several dVAE models on the CelebA dataset [20], with the goal of selecting one as an encoding model of face-selective responses in the human brain. Like standard VAEs, these

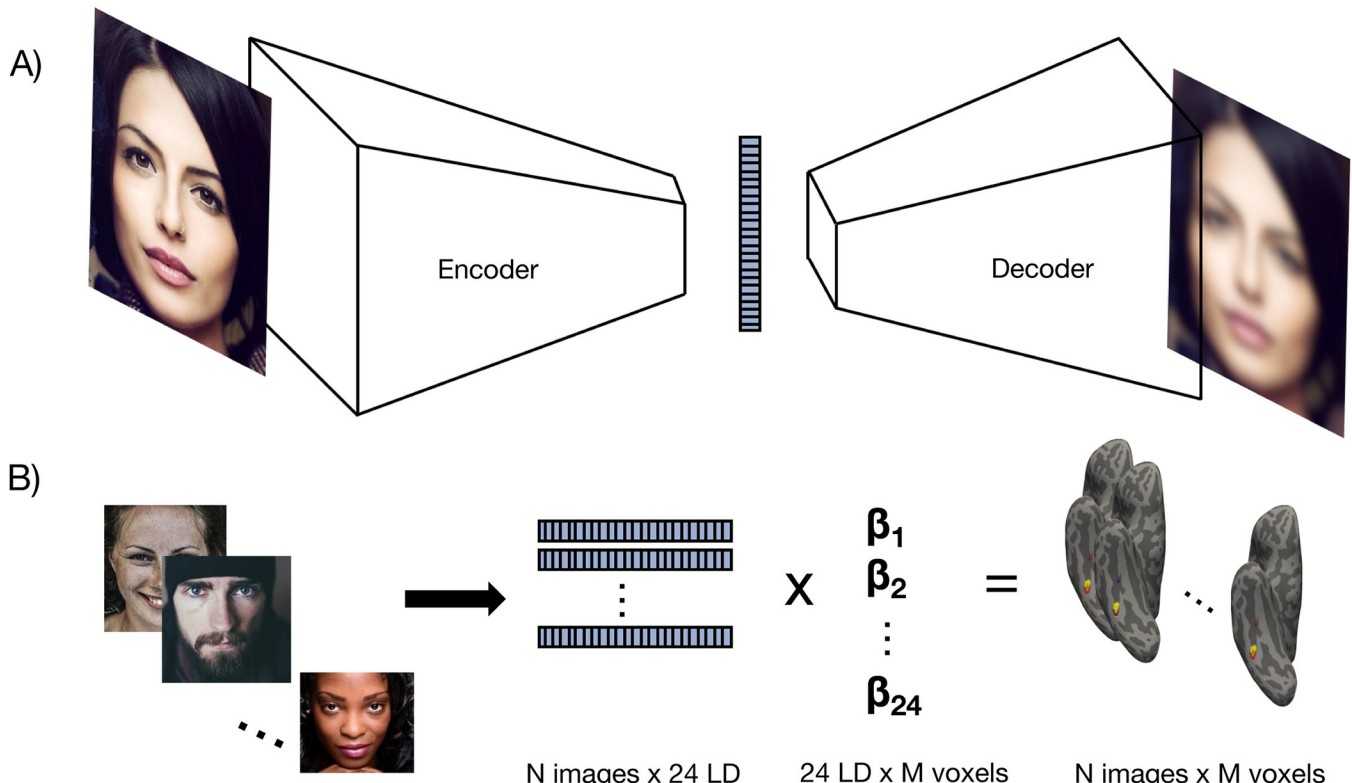

**Fig 1. Model overview and encoding procedure.** A) dVAE model overview. The model takes in a face image (left) and passes it through an encoder, consisting of several convolutional layers (left white trapezoid), to generate a latent vector (blue). Then a decoder (right white trapezoid) uses de-convolution to reconstruct the face image from the latent vector. Like a standard VAE, the dVAE is trained with a cost function to minimize reconstruction error. The dVAE has an additional term to maximize statistical independence (KL-divergence) between elements in the latent vector. B) Encoding procedure. A GLM is used to learn a linear mapping (beta weights) between the latent response to each training image and fMRI responses. At testing, a new test image is passed through the model to extract its 24-dimensional latent vector. This vector is then multiplied by the learned beta weights to generate a predicted voxel response. Because of license restriction, face images are representative of images in the CelebA dataset. Images from top to bottom are cropped from "2150881.png", "woman-1867431_1280.jpg", "1867175.png", and "856125.png" from Pixabay.com and distributed under their content license.

models have an encoder, which transforms an image into a lower-dimensional latent space via convolution, and a decoder, which aims to reconstruct the image from the latent representation (Fig 1A). The models were trained to minimize reconstruction error and had an additional training objective to maximize KL divergence between latent dimensions. Based on a hyperparameter search over previously published model architectures, number of latent dimensions, and model-specific disentanglement parameters to maximize disentanglement (see Methods M1), we selected FactorVAE [16] with 24 latent dimensions as our dVAE model.

After training the dVAE, the authors rated all dimensions by inspecting the faces generated by traversing values of a single latent dimension while keeping all others constant. These latent traversals were often highly interpretable, producing faces that seem to vary along a single dimension, such as facial expression or 3D rotation (Fig 2 and S1–S2 Videos). Out of the 24 latent dimensions, the authors agreed on semantic labels for 16 (14 unanimously and two for a single rater, See Methods M2, Table 1), which included both identity-relevant (dimensions 8–12, 14–16) and irrelevant (dimensions 1–7, 13). These dimensions can also be separated roughly into lower-level visual dimensions that are not face-specific (1–7: lighting, image tone, background, rotation), and face-specific features (8–16), though we note this distinction is not entirely clear cut (see Discussion). The other 8 dimensions were considered entangled,

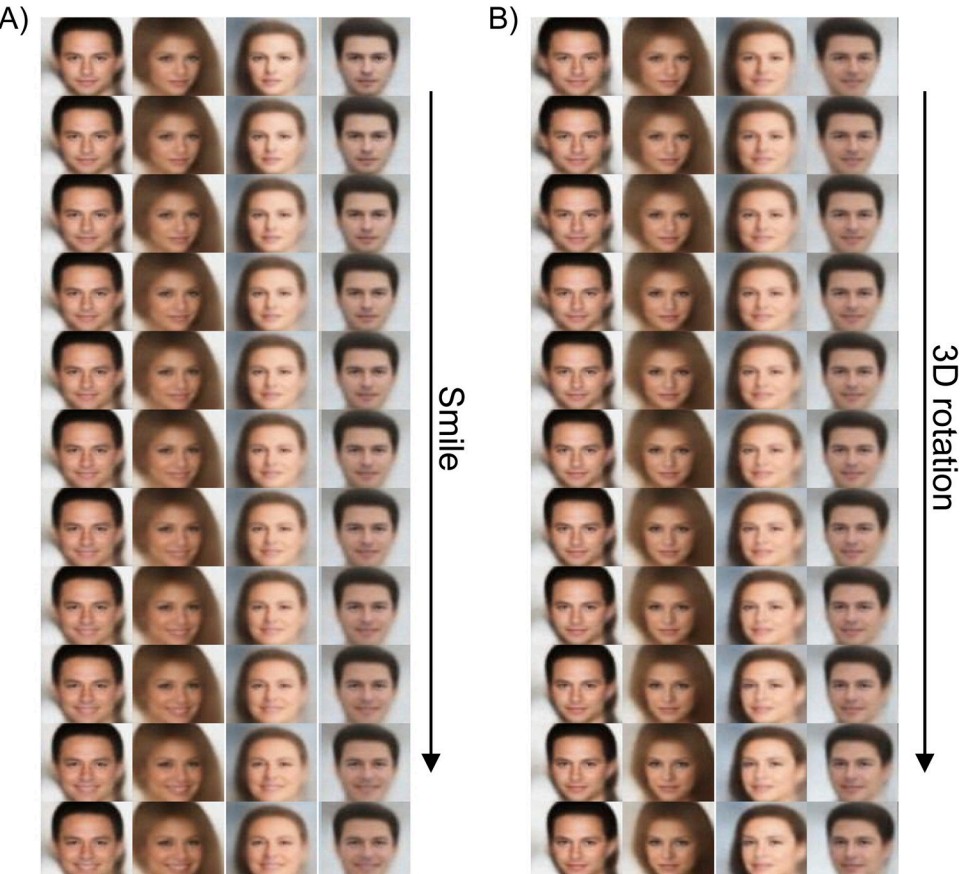

**Fig 2. Latent traversals for two dVAE dimensions.** Images generated by the dVAE when one latent dimension labeled as 'smile' (A) or '3D rotation' (B) is varied from -1 to +1, and other dimensions are held constant. The primary change in the image corresponds to the labeled dimension, suggesting these dimensions are effectively disentangled and highly interpretable. Base images are from the CelebA dataset [20].

containing multiple or uninterpretable transformations. In contrast to the dVAE, traversals from the standard VAE each contained several changing factors (S1–S3 Videos).

We compared our dVAE to two control models. First, we used a standard entangled generative VAE matched in terms of training and hyper-parameters. Second, we used the penultimate layer of a popular DCNN, the discriminatively trained VGG-Face based on VGG16 [21,22]. To match model dimensions, we reduced the dimensionality of the VGG-Face representations to the first 24 principal components, which captured 70.7% of the variance. While the dVAE and VAE latent dimensions shared a similar geometry (CCA r = 0.92), the dVAE and VGG latent spaces were only moderately correlated (CCA r = 0.52), suggesting that discriminative versus generative training frameworks result in different face representations.

**Table 1. Labels assigned to 24 dimensions by human annotators.** Annotators agreed on labels for 16 of 24 dimensions. Colors correspond to plots in Figs 5–6.

| 1 | 2 | 3 | 4 | 5 | 6 | 7 | 8 | 9 | 10 | 11 | 12 | 13 | 14 | 15 | 16 | 17–24 |
|---|---|---|---|---|---|---|---|---|---|---|---|---|---|---|---|---|
| Lighting /Face width | Image tone | Background | Background | 3D rotation/ Lighting | 3D rotation | Elevation | Hair part | Hair | Hair | Hair | Hairline | Smile | Skin tone | Gender appearance | Face width | Entangled |

## Disentangled models provide a good match to ventral face-selective regions

We used a publicly available fMRI dataset [11], where four subjects viewed roughly 8000 face images each a single time. Each subject also viewed 20 face test images between 40–60 times. This approach is in line with recent theoretical and empirical work highlighting the benefits of a small-n, condition-rich design [23,24]. Data were pre-processed and projected onto subjects' individual cortical surfaces. We estimated a linear map between the latent representation of each model and the fMRI data via a generalized linear model (GLM) on the training data (Fig 1B). To predict fMRI responses to each held out test image, we extracted the latent representation for that test image from each model and multiplied them by the linear mapping learned in the GLM.

We evaluated encoding performance for three face-selective ROIs, the Fusiform Face Area (FFA), Occipital Face Area (OFA), and posterior Superior Temporal Sulcus (pSTS), as well as face-selective voxels across the whole brain, identified in a separate face-object localizer experiment (see Methods M4). Despite the additional disentanglement constraint, the dVAE model achieves similar encoding performance to the standard VAE and VGG in FFA and OFA (Fig 3

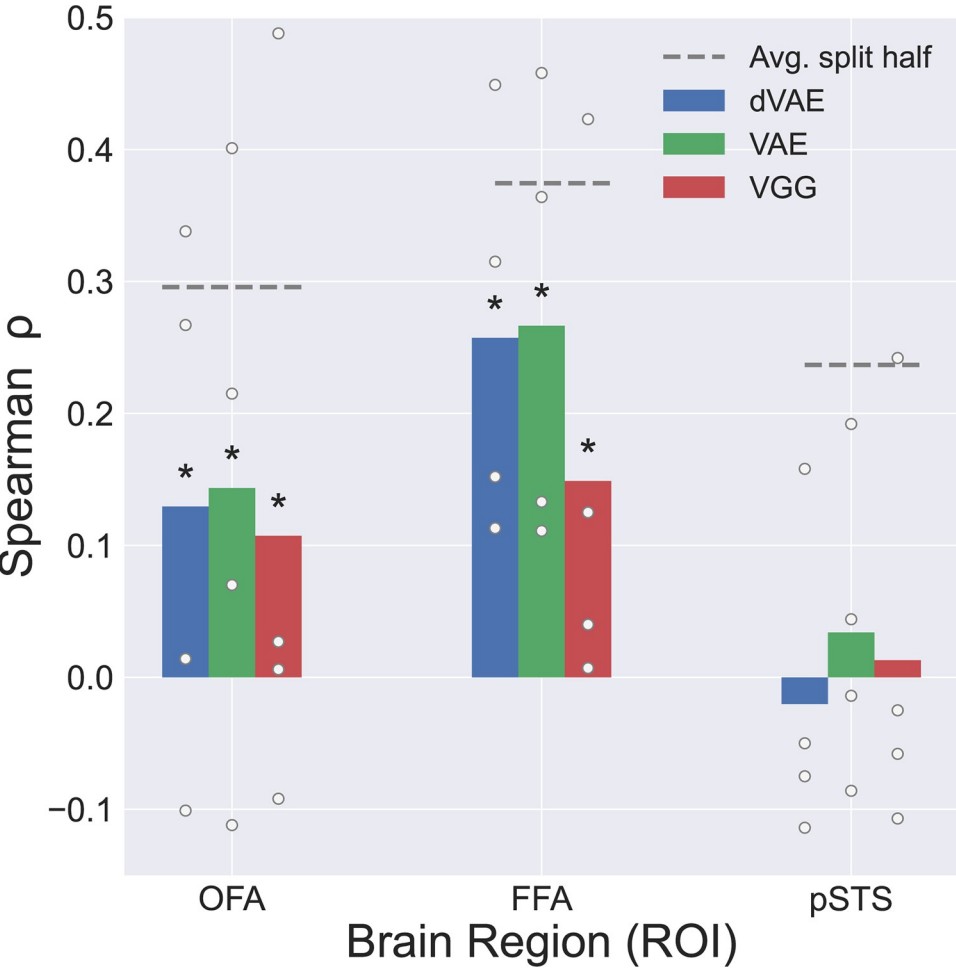

**Fig 3. Encoding performance by ROI.** Average correlation between model prediction and true fMRI responses on held out test images for dVAE (blue), VAE (green), and VGG (red). Dots represent individual subject performance. Asterisks represent significant (p<0.05) results at the group level based on permutation test. Dashed line is average split half reliability across subjects on test image responses.

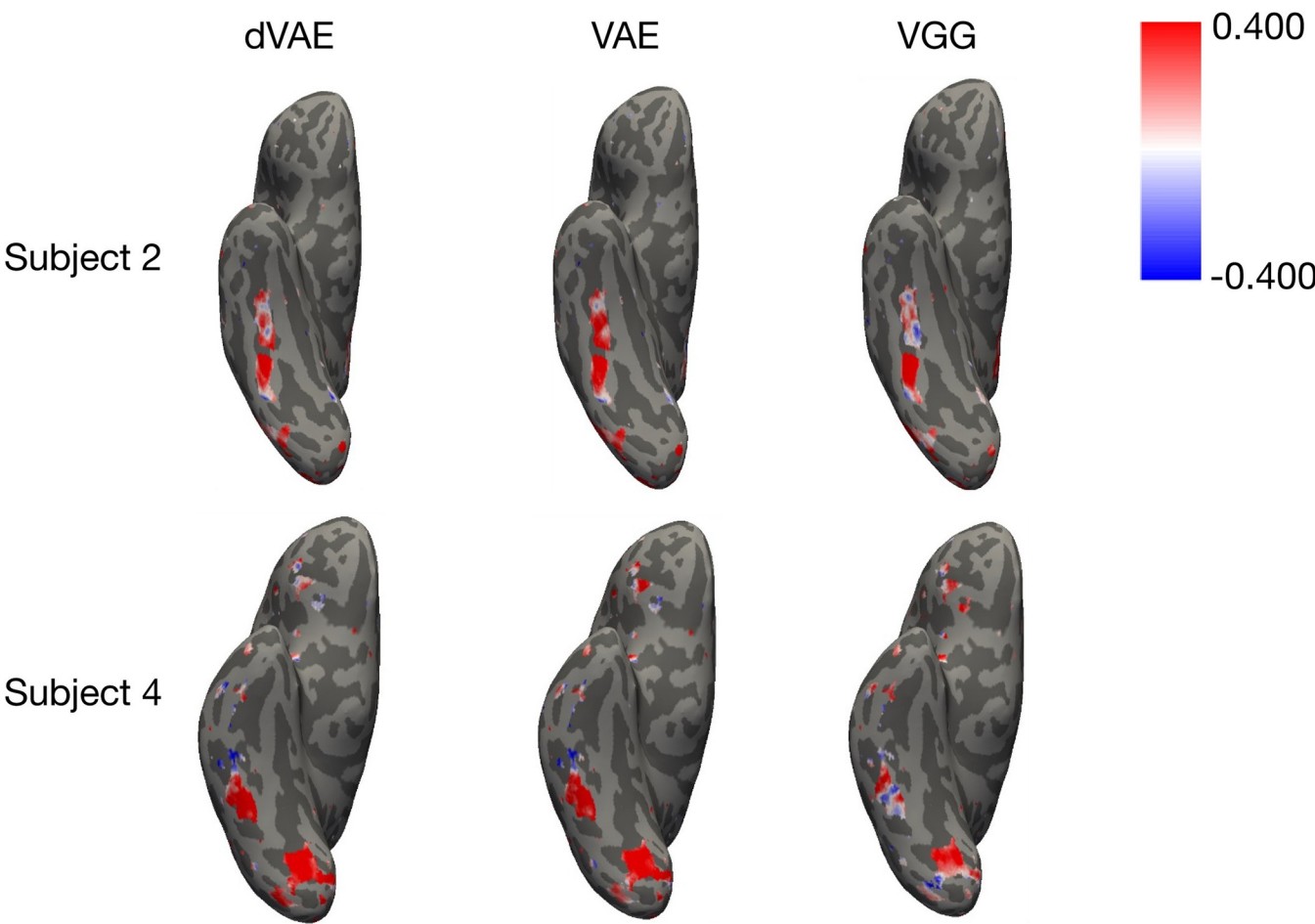

**Fig 4. Whole brain encoding.** Encoding model performance in all face-selective voxels. Ventral view of two representative subjects for dVAE (left), VAE (center) and VGG-face (right).

and S1 Table). At the group level, all models perform significantly above chance (p<0.001) in OFA and FFA. Additionally, both dVAE and VAE have significantly higher predictivity than VGG in the OFA and FFA at the group level (S1 Table). The models also performed similarly across all face-selective voxels in the brain (Figs 4, S1, and S2). None of the models provided consistently above chance accuracy in pSTS (Fig 3), perhaps due to the fact that all stimuli were static faces and lateral face regions have been shown to be selective for dynamic stimuli [4].

**Higher-level, identity-relevant dimensions are represented in more anterior face-selective regions.** The main advantage of disentangled encoding models is the ability to examine how voxels respond to semantically meaningful dimensions. To do this, we performed preference mapping by predicting fMRI responses based on the dVAE latent vector and learned beta weights for each individual latent dimension. Preference mapping is similar to directly comparing the learned beta weights for each feature, but more robust since it is done on held out test data, and more interpretable since the outputs are bounded correlation values versus arbitrarily scaled beta weights [25]. High predictivity of a particular dimension in a particular brain region indicates that changes along that dimension predict changes in neural activity and does not necessarily mean that specific region codes for or is selective to that dimension.

We first performed preference mapping within each ROI (Fig 5). In the OFA, two dimensions, lighting/face width and image tone, were significantly predictive at the group level.

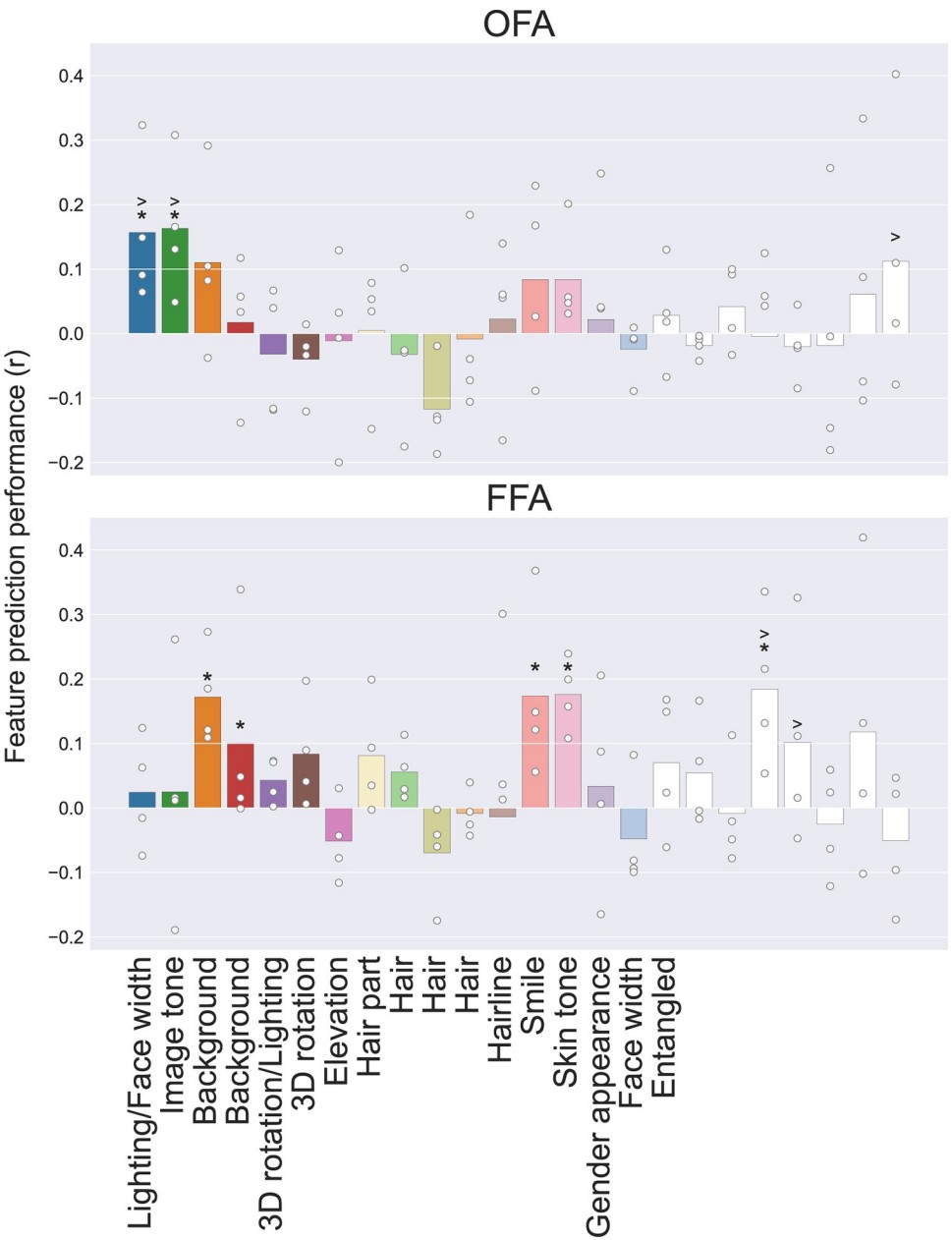

**Fig 5. Preference mapping results for each dimension and ROI.** Average feature predictivity across subjects is shown for each individual dimension. Dots represent individual subjects. Asterisks represents significance versus chance and > represents significantly greater response in one ROI versus the other (p < .05) at the group level. Dimensions are colored according to Table 1.

These dimensions include lower-level visual changes, which are not face-specific, though we note the first dimension also contains information about face width, which may be predictive of identity (see Discussion). In FFA, one lower-level dimension (background) was significantly predictive at the group level. Additionally, two higher-level, face-specific dimensions, smile and skin tone, and one entangled dimension were also significantly predictive of FFA voxel responses at the group level. When comparing the two ROIs, OFA was significantly better predicted by lighting, image tone, and one entangled dimension, and FFA was significantly better

predicted by two entangled dimensions. These FFA-dimensions included both identity-specific features like skin tone and changeable aspects of faces like expression. As with the full model performance in the STS, the performance from most individual dimensions is also worse than the other ROIs (S3 Fig).

To understand how dimensions are represented across the brain, we can visualize their predictivity in a winner take all manner on the surface of the brain (S4 Fig). Similar to the ROI analysis, most posterior voxels were best predicted by image-level changes in background and lighting. More anterior regions, including FFA, and in some subjects, anterior temporal lobe (ATL) regions not included in our ROI analysis, also showed responses for face-specific dimensions like smile (light red) and identity-relevant dimensions like skin tone (light pink), hairstyle (light green, light yellow and light orange), and gender appearance (light purple). Some subjects also show anterior ventral voxels best predicted by visual features like background (dark orange). Interestingly, entangled dimensions (white) were predictive in face-selective voxels throughout the cortex.

**Disentangled models isolate identity relevant face information.** Another benefit of disentangled encoding models is the ability to study and group dimensions based on semantically meaningful attributes. One particularly important distinction for face processing is the separation of identity relevant factors (e.g., gender appearance, skin tone, and face shape) from identity irrelevant factors (e.g. lighting, viewpoint, and background). We decoded identity from our 20 test images using different subsets of dimensions: identity-relevant, identity-irrelevant, and entangled. Note that in our set, identity-relevant dimensions include all face-specific features identified above, with the exception of smile, which is not relevant to identity. Identity-relevant dimensions provided the highest identity decoding accuracy, almost equal to using all dimensions, whereas identity-irrelevant dimensions had the lowest, providing proof of concept that distinctions between our disentangled dimensions capture meaningful semantic information (Fig 6).

The role of information contained in the remaining entangled dimensions of a disentangled model is an open question in AI, so we next sought to examine the extent of identity information in these dimensions. The entangled dimensions contained some identity information as illustrated by their above-chance decoding. However, entangled dimensions do not appear to capture information beyond the identity-relevant dimensions as shown by the similar decoding performances of identity-relevant features and the combination of identity-relevant and entangled features.

## Discussion

We introduced a novel encoding framework for interpreting human fMRI data. Our method allows us to identify semantically meaningful dimensions in an unsupervised manner from large datasets. This disentanglement improves interpretability without a large degradation in encoding performance, as seen by the similar performance between the disentangled and standard VAEs. Our results also suggest that low- and high-level properties are represented in posterior versus anterior brain regions, consistent with prior data and models of face processing [2,26–28]. Here we operationalized low versus high-level dimensions as those that are general visual changes (dimensions 1–7) versus face-specific changes (dimensions 8–16), though these are not necessarily identity-relevant. We note though that this distinction is not entirely clear cut as the 3D properties of faces are unique, and it is an open question whether the learned representations for changes in lighting or 3D rotation would generalize to other objects. In addition, properties like skin tone or face-width could also be considered low-level as they affect the overall pixel-level properties of the image. While we identified several identity-relevant

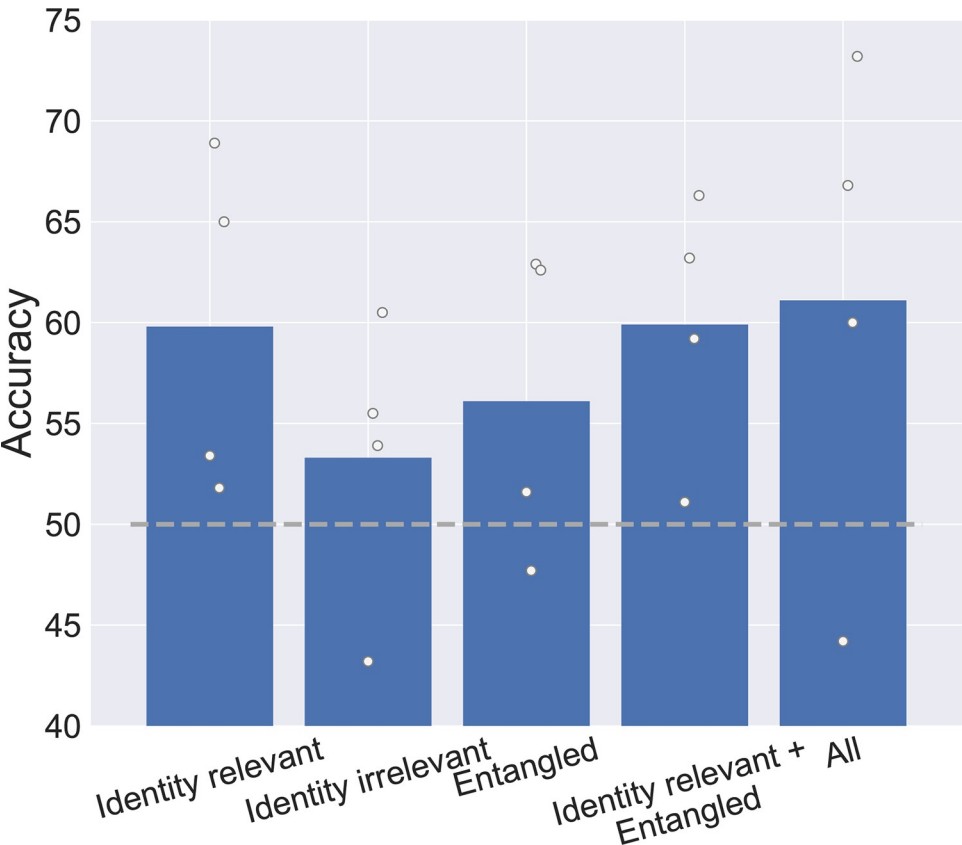

**Fig 6. Facial identity decoding.** Pairwise decoding of face identity from different subsets of dimensions: 8 identity relevant dimensions, 8 identity irrelevant dimensions, 8 entangled dimensions, the combination of identity relevant with entangled dimensions, as well as all dimensions. Mean values shown as blue bars with individual subjects shown as dots. The dashed line at 50% indicates the level of chance performance.

dimensions in FFA, consistent with prior work [28,29], we also found sensitivity to several changeable aspects of faces, including expression, in FFA and other ventral face-selective voxels. These results add to mounting evidence [2] challenging the idea of a clear-cut distinction between identity and expression coding in ventral and lateral face regions [26,30,31]. We note, however, that these conclusions are largely based on analyses in OFA and FFA due to low reliability in lateral and anterior regions. An interesting question for future work is how disentangled model models match representations in the extended face processing network.

In addition to improving our understanding of the human brain, this work also yields new insights into representations learned in disentangled models. Through decoding analysis, we showed that the disentangled identity-relevant dimensions contain almost all the face identity information in the fMRI signals, providing novel support for disentanglement in these models. The nature of learned representations in models trained on naturalistic data is an open question in AI [32]. Our approach allows us to investigate the content contained in the remaining entangled dimensions of the dVAE. We found that these entangled dimensions are represented in face-selective regions, and showed for the first time that they contain identity-relevant information, providing new insight into their computational role.

Prior work has found that DCNNs trained for facial identity discrimination only capture a small amount of variance in human face selective regions [33] and do not replicate activity in the primate face patch hierarchy or human behavioral responses [34]. We see an advantage of

our generatively trained encoding models versus the discriminatively trained DCNN particularly in the FFA, although this is not significant in all individual subjects (Fig 3). Interestingly, recent work [33,35] has shown that object-trained networks do a better job of matching human neural responses to faces than face-trained discriminative networks like VGG-Face tested here, though in general face-selective responses are not as well explained by DCNNs as object- and scene-selective regions. It is possible that this is due to richer training datasets available for objects than faces [36], which may lead to higher latent dimensionality [37] and improve model match to visual cortex [38]. To date no disentangled models have been successfully trained on such large and diverse datasets. For the fairest comparison, we focused on models trained only on face images. As disentangled models improve at capturing image variation at larger scales, future work can compare disentangled and non-disentangled models on richer, more varied datasets.

The original paper presenting this fMRI dataset also found good decoding performance across the brain with a generative VAE, achieving much higher decoding performance than the results presented here [11], as have other studies comparing generatively trained neural networks to visual brain responses [39–41]. The original study focused on maximizing fMRI reconstruction and decoding with a high dimensional network (1024 dimensions vs. our 24). We chose our model to have the highest disentanglement which yielded the lowest dimensional network from all those tested in our hyperparameter search (see Methods M1), likely because enforcing statistical independence between latent dimensions via regularization during training becomes less effective as dimensionality increases. Thus our 24-dimensional network is less expressive than higher dimensional networks because it has a much smaller bottleneck for modeling the data distribution. However, the added interpretability afforded by disentanglement allows more fine-grained interpretability of the fMRI data not possible with standard models. Future work should investigate how to combine the interpretability benefits of disentangled models with the expressiveness of high dimensional networks.

Another recent approach has sought to learn disentangled latent representations in a supervised manner [34]. They learn a model which inverts a 3D face graphics program by supervising intermediate representations to match the primitives defined in the program (e.g. 3D shape, texture, and lighting). They find that this network matches primate face representations better than identity trained networks. Importantly, these intermediate representations are prespecified and need to be learned from labeled synthetic data. Many of these prespecified dimensions match those learned by our dVAE, providing further support for disentangled learning as a method to learn relevant latent dimensions in an unsupervised manner. On the other hand, the learned dimensions do not seem to represent fine-grained facial features, like the relative position and size of the eyes or nose, which prior work has found to be represented in the primate face patch system [42]. While it is difficult to say whether this is due to the specific model or dataset we used, it adds to mounting work [43,44] suggesting a more holistic face-based feature coding.

One recent prior study has investigated the correspondence between dVAEs and single neurons in macaque IT [43]. They find several IT neurons that show high one-to-one match with single units in their dVAE. They also demonstrate a high degree of disentanglement in the macaque neurons by showing a strong correlation between model disentanglement and alignment with IT neurons. It is worth noting that only a handful of neurons in the macaque data show high alignment with single disentangled dimensions. Perhaps unsurprisingly given the lower spatial resolution of fMRI, we do not see the same high disentanglement in our data as evidenced by the fact that each region is well predicted by multiple latent dimensions. Even at the voxel level, it may not be possible to see evidence for single disentangled dimensions. It remains an open question whether, and at what spatial scale, the primate face network is

disentangled or shows exact correspondence to the dimensions learned by dVAEs across the visual system.

The content of the disentangled dimensions learned by our dVAE, and all other disentangled models, reflects the distribution of features in its training set. CelebA is a dataset of celebrity images which does not reflect the underlying distribution of faces that people see in daily life. In particular, CelebA faces tend to be young adults, white, and smiling at a camera. One example of how this can affect learned representations can be seen in the smile dimension, which is sometimes entangled with wearing sunglasses (S1 Video), likely reflecting a bias in CelebA that people wearing sunglasses tend to be smiling. More critically, the visual as well as racial and ethnic biases in the dataset likely impact the quality of the learned dimensions [45] and the model's generalization to other datasets. Training models on a more ecologically valid dataset may improve encoding performance by better reflecting the statistics of real-world visual experience.

This work has important applications for cognitive neuroscientists to understand the relationship between semantic factors and neural activity using natural datasets without labels, in a scalable manner. Successfully scaling these models to larger, less constrained datasets is an important, ongoing research challenge. While our current results show some inter-subject variability, as the quality of models and fMRI data increase, our method can be used to identify new semantically meaningful data dimensions with higher precision. Disentangled models have been created for various visual domains including object and scene processing [46–48] and can in theory be applied to any large scale visual dataset. While disentangled models are an active area of research in AI, there has been little investigation of their cognitive and neural plausibility. Our work sheds light on the role of entangled and disentangled dimensions in face representations in the brain and provides avenues for follow-up questions pertaining to their role in identity decoding. Understanding the neural coding of disentangled dimensions in the brain can help inspire novel data representations in AI systems.

## Methods

### M1. Neural Net architecture and training

We trained our VAE models using the TensorFlow DisentanglementLib package [49]. To identify the best disentangled model for our fMRI analyses, we performed a hyperparameter search over model architectures (including beta-VAE [17] and FactorVAE [16]), number of latent dimensions (24, 32, 48, and 64), and architecture-specific disentanglement parameters (beta-VAE $\beta \, \epsilon$ [1,2,4,6,8,16], FactorVAE $\gamma \, \epsilon$ [0, 10, 20, 30, 40, 50, 100] where $\gamma$ = 0 is the same model as a beta-VAE with $\beta$ = 1, a standard VAE). These hyperparameters were selected based on prior work [49]. For every hyperparameter combination, we performed 10 random initializations. This resulted in 240 FactorVAE models (we used the $\beta$ = 1 beta-VAE as our $\gamma$ = 0 FactorVAE instead of training new models) and 240 beta-VAE models. We used beta-VAE without disentanglement (beta = 1) for the standard, non-disentangled VAE models. After training, models were evaluated using the unsupervised disentanglement metric (UDR) [50]. We selected the model with the highest disentanglement score, a FactorVAE model with 24 latent dimensions and $\gamma$ = 10, for subsequent encoding analyses. Of the dimension-matched standard VAE models, we selected the randomly initialized model with the highest disentanglement score as our baseline.

For our baseline discriminative model, we used VGG-Faces [22], a network that uses the VGG architecture [21] and is trained from scratch on 2.6 million face images to predict face identity. To facilitate model comparison, we take the representations at the final fully connected layer and use Principal Component Analysis to reduce the dimensionality to match that of the VAEs.

## M2. Dimension annotation

After training and selecting our disentangled model, we passed 15 face images, not included in training, to the model. For each face image, we generated a set of "traversal images" by changing the value of a single latent dimension (e.g., Fig 2) from -2 to +2. The traversal images for each latent dimension were combined into an animated gif. The two authors labeled each dimension in each gif. We first consolidated the annotations for each annotator across images for each dimension by tallying the labels across the 15 face images (see S2 Table). We then selected labels where both annotators agreed on the majority of images for our final labels (Table 1). The annotators agreed on 14 out of the 16 labeled dimensions. For the two dimensions that the annotators did not agree on, one annotator assigned a majority label and the other did not. In these two cases (dimension 7 head elevation and dimension 11 hair), we assigned the majority label from one annotator. The remaining 8 dimensions were either not labeled or were not labeled consistently.

## M3. fMRI data and preprocessing

We used publicly available fMRI data of four subjects from [11]. Subjects viewed around 8000 "training" face images each presented once, and 20 "test" face images presented between 40–60 times each. Face images were selected at random from the CelebA dataset and passed through a VAE-GAN. Each face was on the screen for 1s followed by a 2s ISI. The experiment was split over eight scan sessions. Subjects were also scanned on 8–10 separate face-object localizer runs to identify face-selective voxels. Data were collected on a Philips 3T ACHEIVA scanner. Subjects provided informed consent and all experiments were conducted in accordance with Comité de Protection des Personnes standards. For more details, refer to the original paper.

Data were pre-processed and projected onto subjects' individual cortical surfaces using Freesurfer [51]. Preprocessing consisted of motion correcting each functional run, aligning it to each subject's anatomical volume and then resampling to each subject's high-density surface. After alignment, data were smoothed using a 5 mm FWHM Gaussian kernel. All individual analyses were performed on each subject's native surface.

## M4. ROI definition

Regions of interest were defined using a group-constrained subject-specific approach [52]. The regions we investigated were the right Fusiform Face Area (FFA), Occipital Face Area (OFA), and Superior Temporal Sulcus (STS). To define our regions of interest (ROIs), we used the published group parcels from [52].

We selected the top 10% of voxels in each parcel using a metric that combined both face-selectivity and reliability on the test data. We first calculated face-selectivity based on face-object localizer runs, and z-scored each subjects' face>object p-values within each parcel to yield a selectivity score $v_s$ for each voxel. We next calculated the split-half reliability in our test data (Spearman r), and z-scored these values within each parcel to generate a reliability score $v_r$. We then summed the normalized selectivity and reliability scores to yield our final selection metric ($v = v_s + v_r$). We restricted our ROI analyses to the right hemisphere because of more selective face responses and increased reliability in our test data. Across the subjects, the FFA had roughly 170 voxels, the OFA 110 voxels, and the STS 170 voxels. For our whole brain analyses, we computed the above metric ($v = v_s + v_r$) for each cortical voxel. We then selected all voxels that scored more than 1.5 standard deviations above the mean.

## M5. Encoding model procedure

We estimated a linear map between the latent dimensions in our models and the fMRI data via a generalized linear model (GLM), following the procedure in the original study [11]. Since each training face image was shown only once, the latent values for that image (rather than the image itself) were included as weighted regressors to increase reliability of the learned beta weights. The latent values for each training face image, the test faces, and a general face "bias" term were all included as regressors, as well as well as nuisance regressors for linear drift removal and motion correction (x, y, z) per run.

To test the accuracy of the encoding model, we extracted the latent dimensions for each test image and multiplied this by the beta weights learned in the GLM and adding the above "bias" term to get a predicted voxel response to each test image (Fig 1). This produced an estimated brain response for each test image. We then compared the predicted fMRI response in each voxel activity to the true voxel activity across all test images using Spearman correlation.

## M6. Preference mapping

To understand the contribution of each latent dimension to brain responses, we followed the same encoding model training procedure described above. In model testing, we then generated the voxel prediction using a single latent dimension value instead of all the latent dimension values and calculated the correlation between the single dimension's predictions and ground truth. We calculated the average prediction for each latent variable within each ROI (Fig 5). For whole brain analyses, we performed preference mapping [25], assigning each voxel's preference label as the dimension which yielded the highest prediction.

## M7. Identity decoding

To understand the identity-relevant information in different latent dimensions we performed identity decoding of our test images. To decode identity, we took the same learned betas ($W$) from the encoding training procedure and multiplied the test fMRI data ($y$) by its pseudo-inverse ($\hat{x} = W^{-1} * (y - b)$) where $b$ is the face bias term. This generated a predicted set of latent dimensions for each test image. We correlated the predicted latent dimensions with the true test latent dimensions and one random foil to assess the pairwise accuracy of the decoding. If the correlation between the predicted latent dimensions and the true latent dimensions was larger than the correlation between the predicted latent dimensions and the foil, that indicated that the identity was correctly decoded. This pairwise comparison was repeated with every test image as the ground truth, and every other test image as the foil for a total of 20*19 = 380 samples per subject. The total accuracy was the number of correct identity decoding divided by 380. We repeated this for different subsets of latent dimensions: all those labeled as identity-relevant (including hair as it offers important cues to facial identity and prior work has shown sensitivity in face-selective voxels [42]), identity-irrelevant, and entangled dimensions.

## M8. Statistical testing

As the underlying distribution of our data was unknown, we used non-parametric, resampling-based statistics. To evaluate whether each model achieved above chance performance, we generated null hypotheses by repeating the above prediction correlations with shuffled test image labels 1000 resample runs. We performed shuffling within subject, and then computed p-values for each individual as well as group-average prediction.

To compare models, we take the difference in prediction and compare this to a null distribution with shuffled model labels. We generated 1000 resample runs and calculate p-values for each two-tailed pairwise model comparison at the individual and group levels (S1 Table).

We followed the above procedure to assess the significance of our preference mapping results to evaluate whether each feature was significantly predictive in each ROI. To compare predictivity of OFA and FFA, we compared their difference in prediction to a shuffled baseline.

## M9. Analysis code

The code for the analysis is available at https://github.com/psoulos/disentangle-faces.

## Supporting information

**S1 Table. Group and individual subject model versus model significance results.** P-value are in parentheses.
(XLSX)

**S2 Table. Tallies for labels assigned to each of fifteen individual images in the labeling experiments by two raters, and each raters' consensus label.**
(CSV)

**S1 Video. Animated latent traversals for all 24 latent dVAE dimensions for one example rendered face.** Dimensions are varied from -2 to +2, with all other dimensions held constant. Images are model generated. Base image is from the CelebA dataset [20].
(GIF)

**S2 Video. Animated latent traversals for all 24 latent dVAE dimensions for a second example rendered face.** Dimensions are varied from -2 to +2, with all other dimensions held constant. Images are model generated. Base image is from the CelebA dataset [20].
(GIF)

**S3 Video. Animated latent traversals for all 24 latent VAE dimensions for the same exampled rendered face in S1 Video.** Dimensions are varied from -2 to + 2 with all other dimensions held constant. Images are model generated. Base image is from the CelebA dataset [20].
(GIF)

**S4 Video. Animated latent traversals for all 24 latent VAE dimensions for the same exampled rendered face in S2 Video.** Dimensions are varied from -2 to + 2 with all other dimensions held constant. Images are model generated. Base image is from the CelebA dataset [20].
(GIF)

**S1 Fig. Whole brain encoding.** Encoding model performance in all face-selective voxels. Ventral view of the remaining two subjects for dVAE (left), VAE (center) and VGG-face (right).
(EPS)

**S2 Fig. Whole brain correlations for all 4 lateral view.**
(EPS)

**S3 Fig. Preference mapping for each dimension in the STS.** Average feature predictivity across subjects is shown for each individual dimension. Dots represent individual subjects. Asterisks represents significance ($p < .05$). Dimensions are colored according to Table 1.
(EPS)

**S4 Fig. Whole brain preference maps.** Each voxel is shaded based on the latent dimension that provides the highest predictivity.
(EPS)

## Acknowledgments

We thank Michael Bonner for helpful discussions on this work, Emalie McMahon and Raj Magesh for feedback on the manuscript.

## Author Contributions

**Conceptualization:** Paul Soulos, Leyla Isik.

**Data curation:** Paul Soulos.

**Formal analysis:** Paul Soulos.

**Funding acquisition:** Leyla Isik.

**Investigation:** Paul Soulos, Leyla Isik.

**Methodology:** Paul Soulos, Leyla Isik.

**Project administration:** Paul Soulos, Leyla Isik.

**Resources:** Leyla Isik.

**Software:** Paul Soulos.

**Supervision:** Leyla Isik.

**Visualization:** Paul Soulos, Leyla Isik.

**Writing – original draft:** Paul Soulos, Leyla Isik.

**Writing – review & editing:** Paul Soulos, Leyla Isik.

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
