## [Decision Letter · Decision Letter 0]

18 Jun 2023

Dear Dr. Isik,

Thank you very much for submitting your manuscript "Disentangled deep generative models reveal coding principles of the human face processing network" for consideration at PLOS Computational Biology.

Your manuscript was reviewed by members of the editorial board and by two independent reviewers. In light of the reviews (below this email), we would like to invite the resubmission of a significantly-revised version that takes into account the reviewers' comments.

We cannot make any decision about publication until we have seen the revised manuscript and your response to the reviewers' comments. Your revised manuscript is also likely to be sent to reviewers for further evaluation.

Sincerely,

Jean Daunizeau

Academic Editor

PLOS Computational Biology

Lyle Graham

Section Editor

PLOS Computational Biology

Reviewer's Responses to Questions

**Comments to the Authors:**

Reviewer #1: The authors explored disentangled generative neural networks as a model for human face processing. Identifying a Factor VAE through model selection, the authors employed human raters to interpret the 24 model feature dimensions, finding 16 semantically meaningful. The authors then compared the disentangled model with two conventional models—a VAE without disentanglement and a VGG-based face classification network, showing the disentangled VAE (dVAE) to be a competitive encoding model of human fMRI activity. Furthermore, higher-level dVAE features better matched responses in more anterior face-selective regions. The dVAE also enabled identity decoding from fMRI activity, almost entirely through the eight identity-relevant disentangled features.

Developing high-performance yet interpretable models is an important research goal in visual neuroscience. Disentangled models are a promising direction showing emerging evidence to match neural representations uniquely well (e.g., Higgins et al., Nat. Commun. 2021; Whittington et al., ICLR 2023, arXiv:2210.01768). However, several concerns limit the potential impact of the manuscript.

A key promise of disentangled models is more interpretable features. The manuscript provides limited objective evidence for this supposition. Figure 1 shows examples of feature dimension traversals, which the reader must evaluate subjectively. Reasonable viewers may disagree as to how cleanly disentangled the features are. For instance, it looks to me that the “smile” dimension spuriously correlates with jaw width. Table 1 provides a consensus label for each feature; it would be more informative to report the original annotations given to individual images per rater. The raw data, perhaps quantified with a word embedding metric and evaluated for inter-rater consistency, would provide a more granular and rigorous measure of how interpretable the disentangled features are. The interpretability of the dVAE should ideally be compared to non-disentangled models (i.e., the VAE with beta = 1 and VGG-Face PCs) to establish the advantage of using a dVAE. The authors should also detail how they recruited the human raters. While it is not necessarily a problem for the raters to include the author(s), this should be reported.

The authors conclude that the disentangled features constitute a decent encoding model of human fMRI data. However, the encoding performance analysis (Fig. 3) likely unfairly represents conventional classification networks. The final fully connected layer of VGG-Face is considerably worse than its intermediate layers (Jiahui et al., biorxiv 2021). VGG-Face is a worse encoding model than ImageNet-trained VGG, which is, in turn, worse than AlexNet and CORnet (Chang et al., Curr. Biol. 2021), both widely used models. It thus remains unclear whether, or by how much, disentangled models suffer an encoding performance cost relative to conventional, less interpretable models for an (as-yet unquantified) gain in interpretability.

A third major conclusion of the manuscript is a posterior-anterior gradient of low- to high-level feature representations. This claim should be supported by a quantitative summary of the evidence (Fig. 4). While the overall conclusion likely holds, some presumably low-level features (e.g., “background”) are predictive in FFA, and other high-level features (e.g., “smile”) are also predictive in OFA. There is also considerable inter-subject variation. In quantifying the low-to-high level gradient, the authors would also help the reader by unambiguously defining what low- and high-level features signify. The manuscript seems to suggest high-level features as equivalent to identity-specific, “less changeable” features. However, this interpretation is confusing given that the authors claim no spatial segregation between identity-relevant and -irrelevant features. For another example of the present ambiguities, skin tone may be considered either a low-level visual attribute (because it affects luminance semi-globally) or a high-level one (in being identity-relevant).

Finally, although outside the scope of the current analyses, the authors may consider exploring the “alignment” between disentangled feature dimensions and fMRI responses (Higgins et al., 2021). Aside from interpretability, neuron-to-neuron alignment is perhaps the other main promise of disentangled models. This alignment is the main advantage over conventional models found by Higgins et al. (who did not quantify interpretability). While a negative result would be difficult to interpret given the limited spatial resolution of fMRI, a positive finding would add considerably to the study's impact.

Minor concerns:

I am unsure about the significance of analyzing the information in the “entangled” dimensions. A conventional model has only entangled features, which necessarily contain all the identity information available to the model. The entangled features of a dVAE might simply correspond to the entangled dimensions in conventional models. It seems more remarkable (and thus worth more emphasis) that the disentangled, identity-specific features in a dVAE (i.e., a third of all its dimensions) almost completely explain its ability to decode identity from fMRI.

The authors laudably performed an extensive model search. However, the optimal model corresponded to extreme values in the searched parameter ranges—the lowest gamma and fewest feature dimensions. Should this prompt the authors to consider model instances with even lower gamma and fewer dimensions? Moreover, although beta-VAE underperforms Factor VAE in the UDR selection metric, it would still be useful to analyze beta-VAE to facilitate comparison with prior studies.

In Fig. 3, the asterisks look very similar to the gray scatter dots.

On lines 4 and 11, pg. 6, the author used “significantly.” It is unclear whether this refers to statistical significance and, if so, what test was concerned.

Line 42, pg. 6: gender appearance should correspond to dark olive color, not green.

In the Introduction, the authors state that "where […] face identity is represented is still largely unknown" (line 36). The Discussion suggests that identity and expression coding are still understood to be clear-cut (lines 38-40). The authors should moderate both statements in light of the Duchaine & Yovel review the authors also cite. The review presents preponderant evidence that FFA contains both identity and expression information.

The authors used Z-scored p-values to define face-selective ROIs, but p-values do not measure effect size. The raw responses are more appropriate for defining a selectivity index, perhaps with a p-value mask.

Citation [14] (Higgins et al., “beta-VAE […]”) should be dated 2017, not 2022.

Reviewer #2: In the manuscript “Disentangled deep generative models reveal coding principles of the human face processing network”, Soulos and colleagues trained a disentangled deep generative model (dVAE) and used this model as an encoding model for human fMRI data to understand the neural representations of the semantic features of human faces. Using semantically meaningful dimensions from the deep learning models to address the problem of human face representations in the brain is an attractive approach that could integrate AI and neuroscience research. The manuscript is concise and well-written. However, I have reservations about the clarity and robustness of the current results.

Major concerns:

1. The authors trained their model solely on a set of images with limited diversity in face features (CelebA). There's a possibility that the model is biased towards certain dimensions or common image statistics specific to this image set. As numerous face image datasets are readily accessible now, I encourage the authors to broaden their training image set to retrain the model. More importantly, to validate their findings (especially with the neural data), they need to use another deep generative model trained with a different image set.

2. Many current disentangled dimensions are not face-relevant, including identity-relevant features such as hairstyles. Only a few dimensions (e.g., gender appearance, face width) are purely face dimensions. These disentangled dimensions also fail to separate fine features such as the location, distance, or size of the eyes and nose, which are crucial for face recognition and are represented hierarchically in the brain. Thus, the current dVAE provides a simple and coarse model of portrait images that is heavily influenced by prominent face-irrelevant features.

3. The authors assert that "Higher-level, identity-relevant dimensions are represented in more anterior face-selective regions." However, this is not clear to me. The dimensions are mixed, and it's challenging to determine which dimensions are “higher-level” ones. Additionally, this study seems to focus more on the three posterior regions (OFA, FFA, and pSTS), so I'm unclear which regions are considered “anterior” and how these anterior regions were defined. This part of the results needs a clear definition and rigorous analysis to compare the representations between “higher vs lower” level dimensions and “anterior vs posterior” face-selective regions to substantiate the conclusion.

4. This study primarily relies on neural data from four participants, and the inconsistent prediction performance among these participants raises concerns. There are significant inter-subject variations in the results, and with only four data points, it's challenging to gauge the robustness of the results and their generalizability across individuals. I have significant reservations about the small number of participants and strongly recommend the authors to include another dataset or more participants to demonstrate the reliability and generalizability of their findings.

5. I am concerned about the capability of human fMRI data to work with deep neural network models, given that fMRI data lacks the spatial resolution necessary to accurately map the features. The correlations shown in this study are also not particularly high, especially considering the strong effect from non-identity low-level features such as the background. The authors have briefly touched upon this in the discussion, but expanding on this point could help readers understand the limitations of this study.

Minor concerns:

1. The method of calculating accuracies in Figure 6 is unclear. From the methods section, it appears that the decoding was based on the correlations between the predicted and actual latent dimensions . However, it is not clear how the authors converted these correlation values into accuracy values. This process requires further explanation.

2. There are no error bars in any of the plots, possibly due to the large variability across individuals when the study only includes four participants. If more participants can be included, it is recommended that error bars are added to aid understanding of the bar plots.

**Have the authors made all data and (if applicable) computational code underlying the findings in their manuscript fully available?**

Reviewer #1: Yes

Reviewer #2: Yes

PLOS authors have the option to publish the peer review history of their article (what does this mean?). If published, this will include your full peer review and any attached files.

Reviewer #1: No

Reviewer #2: No
---

## [Decision Letter · Decision Letter 1]

11 Oct 2023

Dear Dr. Isik,

Thank you very much for submitting your manuscript "Disentangled deep generative models reveal coding principles of the human face processing network" for consideration at PLOS Computational Biology. As with all papers reviewed by the journal, your manuscript was reviewed by members of the editorial board and by several independent reviewers. The reviewers appreciated the attention to an important topic. Based on the reviews, we are likely to accept this manuscript for publication, providing that you modify the manuscript according to the review recommendations.

Sincerely,

Jean Daunizeau

Academic Editor

PLOS Computational Biology

Lyle Graham

Section Editor

PLOS Computational Biology

Reviewer's Responses to Questions

**Comments to the Authors:**

Reviewer #1: I appreciate the revisions the authors made. The results are more straightforward to evaluate and better contextualized.

My main comments this time are to tighten the conclusions. The authors claim the following in the abstract:

1. The majority of learned latent dimensions in [the dVAE] are interpretable by human raters

2. These latent dimensions serve as a good encoding model for human fMRI data

3. [There is] a gradient from low- to high-level face feature representations along posterior to anterior face-selective regions

4. A decoding analysis confirms that the model separates identity-relevant and -irrelevant information

5. [There is] no spatial segregation between identity-relevant and -irrelevant face features

6. The few "entangled" (uninterpretable) dimensions

6a. match responses across the ventral stream

6b. carry significant information about facial identity

I think claims 1, 2, and 4 are now reasonably well supported. I appreciate the more direct description of interpretability ratings and the addition of Table S2. I agree that adding participants for rating is not essential to the paper's more interesting results on brain decoding.

Claim 3 is still unclear. Part of the issue is the wording—I think 'gradient' does not aptly describe comparisons between two ROIs. Moreover, while per-ROI statistics identify two face-specific (i.e., 'high-level') dimensions in FFA, between-ROI statistics only show statistically significant differences in an entangled feature in FFA, which does not directly support the claim of a gradient from low- to high-level features. (I do appreciate the addition of between-ROI statistics.) Another issue is that the evaluation of the voxel-wise results is purely qualitative, and I am unsure to what degree they support the claim of a gradient. Fig. S4 was hard for me to read. A different color map would help, giving high-level and low-level two families of colors (e.g., warm and cold). (The current color map puts features in groups of 2 that are irrelevant to the paper's claims, distracting, and require repeated references to the figure key.)

It is unclear what evidence directly supports claims 5 and 6a. Particularly for claim 5, the word segregation never appears in the main text. Are claims 5 and 6a based on the same evidence as claim 3 (i.e., Figs. 5 and S4)? Are both claims rigorously testable? I.e., what results, ideally quantitative ones, would support or reject the respective claims?

In claim 6b, it again helps to specify whether 'significant' refers to statistical significance. I found no statistical tests associated with Fig. 6. Is claim 6 really about 'above-chance' decoding? It helps to indicate chance in Fig. 6 (50% if I understand correctly).

I think suitably re-wording claims 3, 5, and 6 will not detract from the paper's significance and requires no additional analysis, although additional analysis may further strengthen claim 3.

Minor comments:

The Discussion explains well why the authors chose 24 latent dimensions (lines 10.31-10.40). A preview of this is due when this parameter was first introduced (line 4.31). The number of model latent dimensions is relevant. A different choice can potentially affect the conclusions about interpretability and the two classes of features (identity-relevant or not).

Why do the authors distinguish high- and low-level features vs. identity-relevant and -irrelevant features? The reason is implicit in some places (e.g., Fig. 6) but not in others, and juxtaposing the two categorization systems was confusing (e.g., in the abstract and on page 7). It would help the reader to explain why each analysis used either categorization and emphasize the subtle difference between the two since only one feature distinguishes them (dim 13. 'smile').

Line 4.39, 'agreed on': This phrasing is confusing. The Method (and reviewer response) is unambiguous—the authors agreed on 14 dimensions, and the other two dimensions were interpretable to one rater and conceded by the other.

Line 5.13, 'correlated': Do the latent dimensions correlate or have similar geometry? The word 'correlated' is confusing because it could mean individual features are correlated, which would contradict the result that dVAE features are more disentangled and interpretable.

In Fig. 5, it helps to annotate the dimensions showing significant differences between OFA and FFA.

Lines 7.37–7.40: The posterior voxels are not that clear. Where posteriorly can I see background (oranges) and image tone (pale blue)?

Lines 8.24-26, 'The role of information contained in the remaining entangled dimensions of a disentangled model is an open question in AI': I'm still unconvinced this is a significant question in AI. I must be less familiar with the literature than the authors. Thus, the authors can help by discussing or citing work that discusses why residual entangled dimensions are an interesting and important open question in AI.

Line 9.29, 'he combination': typo.

Lines 10.16–19: Given that the authors have conducted a direct (preliminary) analysis on alignment, it is well to mention it here. I agree with the authors about not including the relevant figures in the reviewer response as supplementary figures, but only because the plots show no interpretable differences, not because the analysis itself is distracting.

Reviewer #2: The authors have successfully addressed most of my questions and concerns. The only two points I would like to discuss with the authors are related to previous major concerns 3 and 5.

1. My understanding remains unclear regarding how the authors define the anterior and posterior regions. I am skeptical about relying solely on a few ventral regions to draw conclusions about the gradient. I would recommend that the authors discuss this limitation in the discussion section.

2. Research involving human fMRI data and deep neural network models appears to produce more inconsistent results when modeling faces compared to other object categories, such as scenes from natural datasets. I would be interested in the authors' perspective on this discrepancy.

**Have the authors made all data and (if applicable) computational code underlying the findings in their manuscript fully available?**

Reviewer #1: **No: **Are the trained model weights tested in the paper shared? The GitHub repo has no README, and I didn't find a link to model weights after skimming the code. If the trained weights cannot be shared, the author should affirm that the shared code is sufficient for others to fully reproduce the study.

Reviewer #2: Yes

PLOS authors have the option to publish the peer review history of their article (what does this mean?). If published, this will include your full peer review and any attached files.

Reviewer #1: No

Reviewer #2: No

Figure Files:

Data Requirements:

Reproducibility:

References:

---

## [Decision Letter · Decision Letter 2]

2 Feb 2024

Dear Dr. Isik,

We are pleased to inform you that your manuscript 'Disentangled deep generative models reveal coding principles of the human face processing network' has been provisionally accepted for publication in PLOS Computational Biology.

Best regards,

Jean Daunizeau

Academic Editor

PLOS Computational Biology

Lyle Graham

Section Editor

PLOS Computational Biology

Reviewer's Responses to Questions

**Comments to the Authors:**

Reviewer #1: The authors have commendably addressed all my concerns. I heartily recommend publication. I only have some minor suggestions and do not need to see the manuscript again.

On page 2, lines 19–21, 27–29, 31: Are the authors presenting the notion that disentangled models learn semantically meaningful dimensions as a known fact or a hypothesis tested in the present study? I thought it was the latter. If so, lines 19–21 and 31 jump the gun a bit.

Page 5, lines 15–18: To my understanding, CCA identifies multiple canonical directions in descending order of the data correlation they capture. I’m guessing that the reported scalar r-values correspond to the top canonical direction per model pair. This may well be an implicit assumption in some literatures, but the authors can clarify the meaning.

Page 10, line 7, ‘model models’: typo.

Reviewer #2: The authors have addressed all my concerns, and I have no further questions. I recommend the article to be published.

**Have the authors made all data and (if applicable) computational code underlying the findings in their manuscript fully available?**

Reviewer #1: Yes

Reviewer #2: Yes

PLOS authors have the option to publish the peer review history of their article (what does this mean?). If published, this will include your full peer review and any attached files.

Reviewer #1: No

Reviewer #2: **Yes: **Guo Jiahui

---

## [Editor Report · Acceptance letter]

19 Feb 2024

PCOMPBIOL-D-23-00246R2 

Disentangled deep generative models reveal coding principles of the human face processing network

Dear Dr Isik,

I am pleased to inform you that your manuscript has been formally accepted for publication in PLOS Computational Biology. Your manuscript is now with our production department and you will be notified of the publication date in due course.

With kind regards,

Bernadett Koltai
